# Transposon dynamics in the emerging oilseed crop *Thlaspi arvense*

**Adrián Contreras-Garrido[1]☉, Dario Galanti[2]☉, Andrea Movilli[1], Claude Becker[3], Oliver Bossdorf[2], Hajk-Georg Drost [4]*, Detlef Weigel [1]***

**1** Department of Molecular Biology, Max Planck Institute for Biology Tübingen, Tübingen, Germany, **2** Plant Evolutionary Ecology, University of Tübingen, Tübingen, Germany, **3** LMU Biocenter, Faculty of Biology, Ludwig Maximilians University Munich, Martinsried, Germany, **4** Computational Biology Group, Max Planck Institute for Biology Tübingen, Tübingen, Germany

☉ These authors contributed equally to this work.
* drost@tue.mpg.de (H-GD), weigel@tue.mpg.de (DW)

**Data Availability Statement:** Code used for analysis and figures can be found at: https://github.com/acontrerasg/Tarvense_transposon_dynamics. Sequencing reads can be found at the European Nucleotide Archive (ENA) under accession number

## Abstract

Genome evolution is partly driven by the mobility of transposable elements (TEs) which often leads to deleterious effects, but their activity can also facilitate genetic novelty and catalyze local adaptation. We explored how the intraspecific diversity of TE polymorphisms might contribute to the broad geographic success and adaptive capacity of the emerging oil crop *Thlaspi arvense* (field pennycress). We classified the TE inventory based on a high-quality genome assembly, estimated the age of retrotransposon TE families and comprehensively assessed their mobilization potential. A survey of 280 accessions from 12 regions across the Northern hemisphere allowed us to quantify over 90,000 TE insertion polymorphisms (TIPs). Their distribution mirrored the genetic differentiation as measured by single nucleotide polymorphisms (SNPs). The number and types of mobile TE families vary substantially across populations, but there are also shared patterns common to all accessions. Ty3/Athila elements are the main drivers of TE diversity in *T. arvense* populations, while a single Ty1/Alesia lineage might be particularly important for transcriptome divergence. The number of retrotransposon TIPs is associated with variation at genes related to epigenetic regulation, including an apparent knockout mutation in *BROMODOMAIN AND ATPase DOMAIN-CONTAINING PROTEIN 1* (*BRAT1*), while DNA transposons are associated with variation at the *HSP19* heat shock protein gene. We propose that the high rate of mobilization activity can be harnessed for targeted gene expression diversification, which may ultimately present a toolbox for the potential use of transposition in breeding and domestication of *T. arvense*.

## Author summary

Transposable elements (TEs) are often considered genomic parasites, but they can also generate phenotypic novelty that helps organisms to adapt to new environments. To understand how TEs might contribute to phenotypic diversity and adaptive potential in the emerging oilseed crop *Thlaspi arvense* (field pennycress), we examined the dynamics

PRJEB62093. See S3 Table for details of the datasets. Datasets were uploaded to Zenodo under the DOI: 10.5281/zenodo.6372331. The workflow was based on custom bash and python scripts available at https://github.com/acontrerasg/Tarvense_transposon_dynamics. All the code for short variants calling, filtering and imputation can be found on GitHub (https://github.com/Dario-Galanti/BinAC_varcalling).

**Funding:** The study was supported by Marie Skłodowska Curie ETN EpiDiverse (EU Horizon 2020 Grant Agreement No. 764965; C.B., O.B., D. W.), the European Research Council (Grant Agreement No. 716823 "FEAR-SAP"; C.B.), the Novo Nordisk Foundation Novozymes Prize and the Max Planck Society (D.W.). The funders had no role in study design, data collection and analysis, decision to publish, or preparation of the manuscript.

**Competing interests:** I have read the journal's policy and the authors of this manuscript have the following competing interests: D.W. holds equity in Computomics, which advises breeders. D.W. advises KWS SE, a plant breeder and seed producer. All the other authors have declared that no competing interests exist.

of TE variation in a geographically diverse sample of this species. By surveying almost 300 wild accessions from North America and Eurasia we discovered over 90,000 polymorphic TE insertions. We identified not only genetic factors that vary between populations and that are associated with TE mobilization, but also TE families that are most likely to generate genetic diversity of interest to breeders.

## Introduction

Transposable elements (TEs) are often neglected, mobile genetic elements that make up large fractions of most eukaryotic genomes [1]. In plants with large genomes, such as wheat, TEs can account for up to 85% of the entire genome [2,3]. Due to their mobility, TEs can significantly shape genome dynamics and thus both long- and short-term genome evolution across the eukaryotic tree of life. TEs are typically present in multiple copies per genome and they are broadly classified based on their replication mechanisms, as copy-and-paste (class I or retrotransposons) or cut-and-paste (class II or DNA transposons) elements. The two categories can be broken down into superfamilies based on the arrangement and function of their open reading frames [4]. Further distinctions can be made based on the phylogenetic relatedness of the TE encoded proteins [5,6]. To minimize the mutagenic effects of TE mobilization, host genomes tightly regulate TE load through an array of epigenetic repressive marks that suppress TE activity [7–9].

While epigenetic silencing of TEs is important for the maintenance of genome integrity and species-specific gene expression, TE mobilization can also generate substantial phenotypic variation through changing the expression of adjacent genes, either due to local epigenetic remodeling or direct effects on transcriptional regulation [10]. Because TE activity is often responsive to environmental stress [11–13] and other environmental factors [14–17], it has been proposed that it could be used for speed-breeding through externally controlled transposition activation [18].

*Thlaspi arvense*, field pennycress, yields large quantities of oil-rich seeds and is emerging as a new high-energy crop for biofuel production [19–21]. As plant-derived biofuels can be a renewable source of energy [22], the past decade has seen efforts to domesticate this species and understand its underlying genetics in the context of seed development and oil production. *Thlaspi arvense* is particularly attractive as a crop because it can be grown as winter cover during the fallow period, protecting the soil from erosion [19]. Natural accessions of *T. arvense* are either summer or winter annuals, with winter annuals being particularly useful as potential cover crop [23]. Native to Eurasia, *T. arvense* was introduced and naturalized mainly in North America [24].

As a member of the Brassicaceae family, *T. arvense* is closely related to the oilseed crops *Brassica rapa* and *Brassica napus*, as well as the undomesticated model plant *Arabidopsis thaliana* [25]. A large proportion of the *T. arvense* genome consists of TEs [26], and TE co-option has been proposed as a mechanism particularly for short-term adaptation and as a source of genetic novelty [27]. As in many other species, differences in TE content is likely to be a major factor for epigenetic variation as well, especially through remodeling of DNA methylation [28].

Here, we use whole-genome resequencing data from 280 geographically diverse *T. arvense* accessions to characterize the inventory of mobile TEs (the 'mobilome'), TE insertion patterns of class I and class II elements and their association with variation in the DNA methylation

landscape. We highlight a small TE family with preference for insertion near genes, which may be particularly useful for identifying new genetic alleles for *T. arvense* domestication.

## Results

### Phylogenetically distinct transposon lineages shape the genome of *T. arvense*

To be able to understand TE dynamics in *Thlaspi arvense*, we first reanalyzed its latest reference genome, MN106-Ref [26]. In total, 423,251 transposable elements were categorized into 1984 unique families and grouped into 14 superfamilies (S1 Table), together constituting 64% of the ~526 Mb MN106-Ref genome. Over half of the genome consists of LTR (Long Terminal Repeat)-TEs. Using the TE model of each LTR family previously generated by structural *de novo* prediction of TEs [26], we assigned 858 (~70%) of the 1,205 Ty1 and Ty3 LTR-TEs to known lineages based on the similarity of their reverse transcriptase domains [5] (Fig 1A).

The most abundant LTR-TE lineage in *T. arvense* is Ty3 Athila (S2 Table) with ~180,000 copies, 10-fold more than the next two most common lineages, Ty3 Tekay (~57,000) and Ty3 CRM (~30,000). The most abundant Ty1 elements belonged to the Ale lineage, with 108 families, while the Alesia and Angela lineages were represented only by one family each (S2 Table).

Next, we compared the genomic distribution of lineages within the same TE superfamily (Fig 1B). In the Ty1 superfamily, CRM showed a strong centromeric preference, whereas Athila was more common in the wider pericentromeric region. In the Ty1 superfamily, Ale elements were enriched in centromeric regions, whereas Alesia showed a preference for gene-rich regions.

### *Thlaspi arvense* LTR retrotransposons present signatures of recent activity

To assess the potential and natural variation of TEs transposition across accessions, we used the complete set of protein domains identified for a respective TE model to classify each family as either potentially autonomous or non-autonomous (METHODS). About 60% of all TE families (1,260 out of 2,038) encoded at least one TE-related protein domain, but only about a quarter had all protein domains necessary for transposition, and we classified these 537 families as autonomous. Autonomous TE families had on average more and longer copies than non-autonomous ones, although both contributed similarly to the total TE load in the genome (S1 Fig). Next, we focused on individual, intact LTR-TE copies, since they are often the source of ongoing mobilization activity (13)(18)(56). Overall, the 193 autonomous LTR-TE families had more members without apparent deletions than the 1,027 non-autonomous LTR-TE families (2,039 versus 339). Intact LTR-TEs from autonomous families tended to be evolutionarily younger and more abundant than their non-autonomous counterparts (Fig 1C). As for lineages, Athila was the lineage with the most intact members, followed by Tekay and CRM (Fig 1D), although estimates of insertion times revealed Ale and Alesia Ty1 lineages as actors of the most recent transposition bursts (Fig 1E).

### TE polymorphisms in a collection of wild *T. arvense* populations

Our analysis of the MN106-Ref reference indicated that a substantial part of the genome consists of autonomous TE families. To learn how TE mobility has shaped genomes at the species level, we surveyed differences in TE content in a large collection of natural accessions. We compiled whole-genome sequences of 280 accessions from different repositories (S3 Table), covering twelve geographic regions, and much of the worldwide distribution of *T. arvense* in its native range and in regions where it has become naturalized (Fig 2A).

We first characterized the population structure of this collection with a subset of high-confidence SNPs and short indels that we used to cluster the accessions by principal component

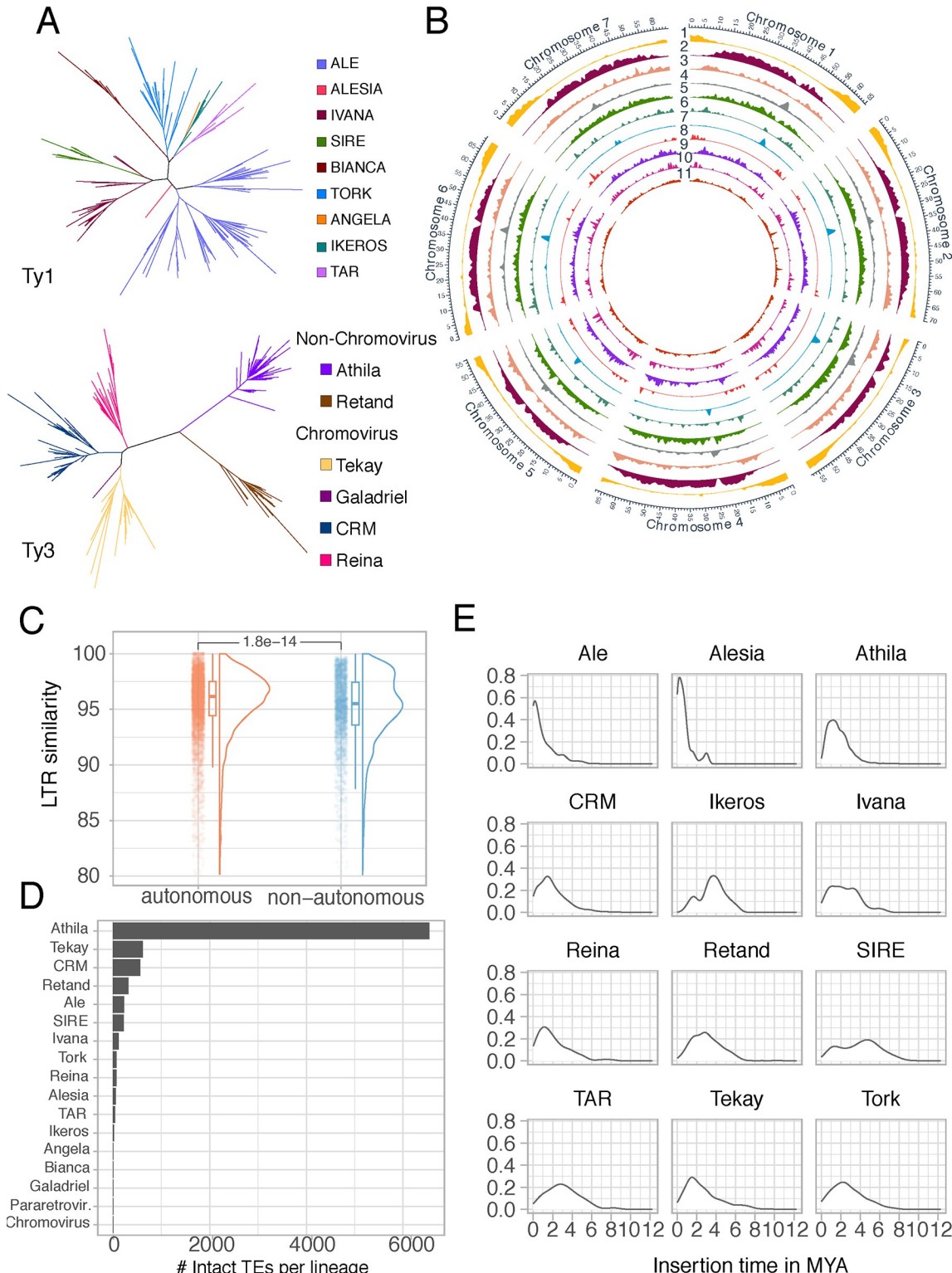

**Fig 1. Genome-wide distribution and classification of TE families and superfamilies in the *T. arvense* reference genome MN106-Ref. (A)** Phylogenetic tree of LTR retrotransposons based on the reverse transcriptase domain. (**B**) Genome-wide distribution of TE family and superfamily abundances. The tracks denote, from the outside to the inside, (1) protein-coding loci, (2) Athila, (3) Retand, (4) CRM, (5) Tekay, (6) Reina, (7) Ale, (8) Alesia, (9) Bianca, (10) Ivana, (11) all DNA TEs. (**C**) Evolutionary age estimates of intact copies of autonomous versus non-autonomous TE families. P-value is computed based on performing a Wilcoxon Rank Sum test.

(**D**) Total number of intact TEs in different lineages. (**E**) Distribution of insertion time estimates for intact LTR elements across different LTR TE lineages (shown if number of intact TEs was greater than 10).

analysis (PCA) (Fig 2B) (Methods). We also constructed a maximum likelihood tree without considering migration flow for these populations, using the two sister species *Eutrema salsugineum* and *Schrenkiella parvula* as an outgroup (S2 Fig). North American accessions clustered together with European accessions, in support of *T. arvense* having been introduced to North America from Europe. Chinese accessions formed a separate cluster, but the most isolated cluster was composed of Armenian accessions, as it has been reported previously [20,26].

Next, we screened our data for TE insertion polymorphisms (TIPs), *i.e.*, TEs not present in the reference genome assembly. This will in most cases be due to insertions that occurred on the phylogenetic branch leading to the non-reference accession, although it formally could also be the result of deletion or excision events of a shared TE on the branch leading to the reference accession.

We detected 18,961 unique insertions, which were unequally distributed among populations, with an excess of singletons (5,617 singletons) (Fig 2C). The allele frequency of TIPs was on average lower than that of SNPs (Fig 2C), with the caveat that detection of TIPs may incur more false negatives. Saturation analysis (Fig 2D) indicated that we were far from sampling the total TE diversity in *T. arvense*, especially in Armenian and Chinese accessions. Taken at face value, the disparity in singleton frequencies between TIPs and SNPs would suggest either that TIPs are on average evolutionarily younger than SNPs, or that there is stronger selection pressure against TE insertions [29] (Fig 2C). What speaks against the latter view is that TIPs in the gene-rich fraction of the genome, near the telomeres, have higher allele frequencies (Fig 2E), while TIPs in the pericentromeric regions are more abundant, but have lower allele frequencies (see S3 Fig for a statistical assessment).

We complemented our analysis of TIPs with a corresponding analysis of TE absence polymorphisms (TAPs), which we define as TEs that are found in the reference assembly but missing from other accessions. This could be due to insertions having occurred on the phylogenetic branch leading to the reference accession or excisions of DNA TEs by a cut-and-paste mechanism. TAPs were detected using a custom TAP annotation pipeline (METHODS).

Overall, a comparison of TIPs and TAPs distributions by PCA showed Armenian accessions to be clear outliers, with all other accessions clustering closely together (Figs 2B and S5), indicating that most of the observed TE variation reflects the population structure observed with SNPs. As with SNPs, Armenian accessions harbor the largest number of both TIPs and TAPs. If we look at the impact of these polymorphisms on the genomic landscape (Fig 3A), we find a major hotspot of TAPs in chromosome 4 for a subset of accessions from Southern Sweden. There also appears to have been major insertion activity in the clade leading to the reference accession, as indicated by the high density of reference insertions missing in all other populations at the ends of chromosomes 4 and 5. For both TIPs and TAPs, the major source of TE polymorphisms comes from activity of Ty3 LTRs (RLGs), especially Ty3 Athila (Fig 3B). Many other TE families contributed to both TIPs and TAPs as well, with 1,203 families having at least one TIP, and 1,268 having at least one TAP. The more distant a population is geographically from the reference, the greater the contribution of non-autonomous families to the TIP load, with the exception of Northern Germany (Fig 3C).

Across all populations, most TE activity was due to a small set of 25 TE families, with the Athila lineage standing out in particular (Fig 3D). For highly active TE families, TIPs were more diverse than TAPs, as the latter were predominantly driven by LTR retrotransposons.

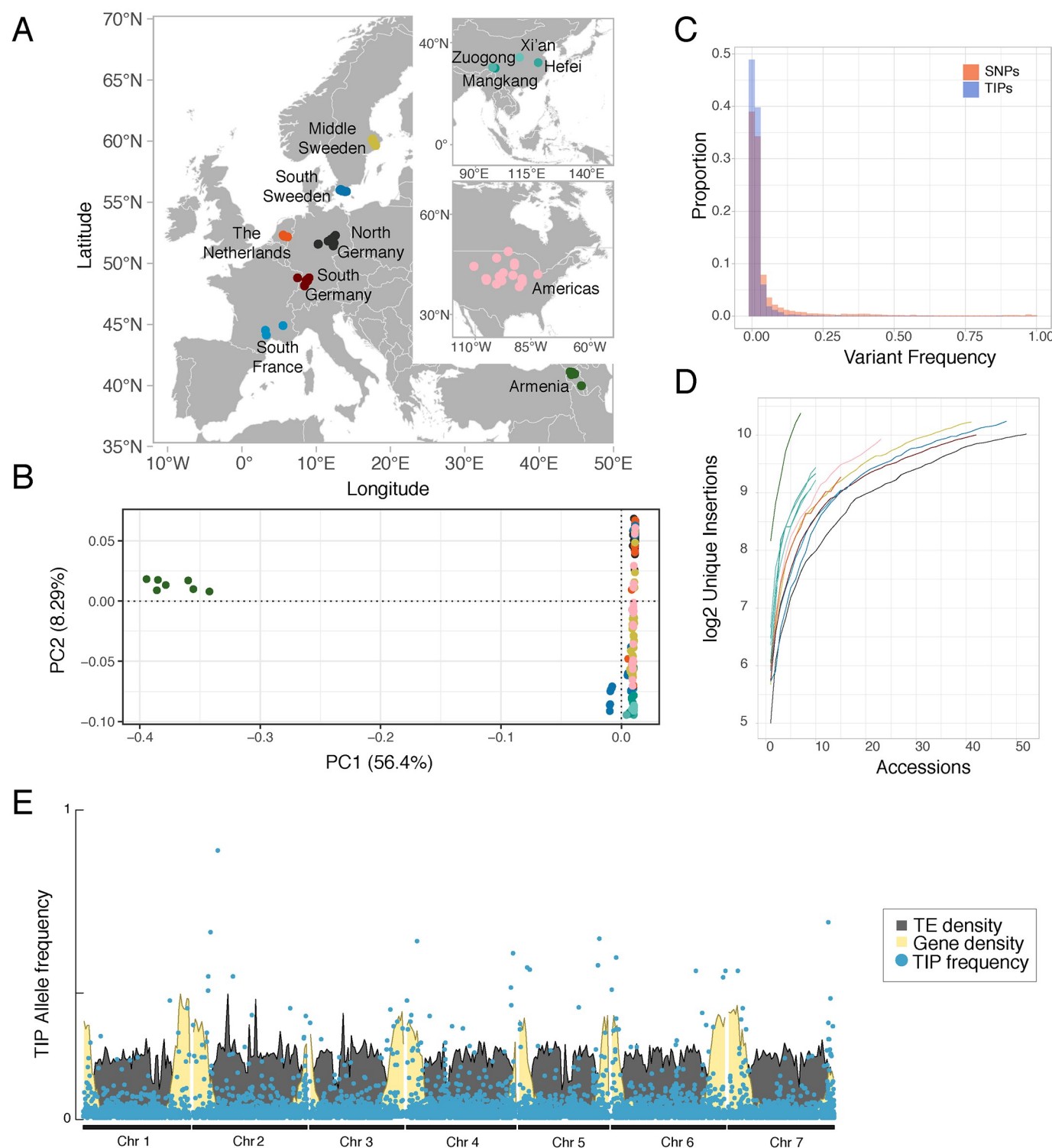

**Fig 2. The genome-wide landscape of TE insertion polymorphisms in *T. arvense*.** (**A**) Distribution of accessions across their native Eurasian and naturalized North American range in the Northern hemisphere (omitting a sample from Chile, included in the Americas group). (**B**) A SNP-based principal component analysis (PCA) of all accessions, with color code as in (A). Due to the fact that the accessions contributing to the Armenian cluster are separated from the other geographic populations, we recalculated a PCA without the Armenian samples as shown in S4 Fig. (**C**) Allele-frequency spectrum of TIPs (blue) and SNPs (red). (**D**) Cumulative sums of unique insertions per region as a function of sampled accessions. (**E**) Average TIP frequencies over 100 kb windows along the genome, compared to gene and TE densities, also displayed in 100 Kb windows. Map source: naturalearthdata.com.

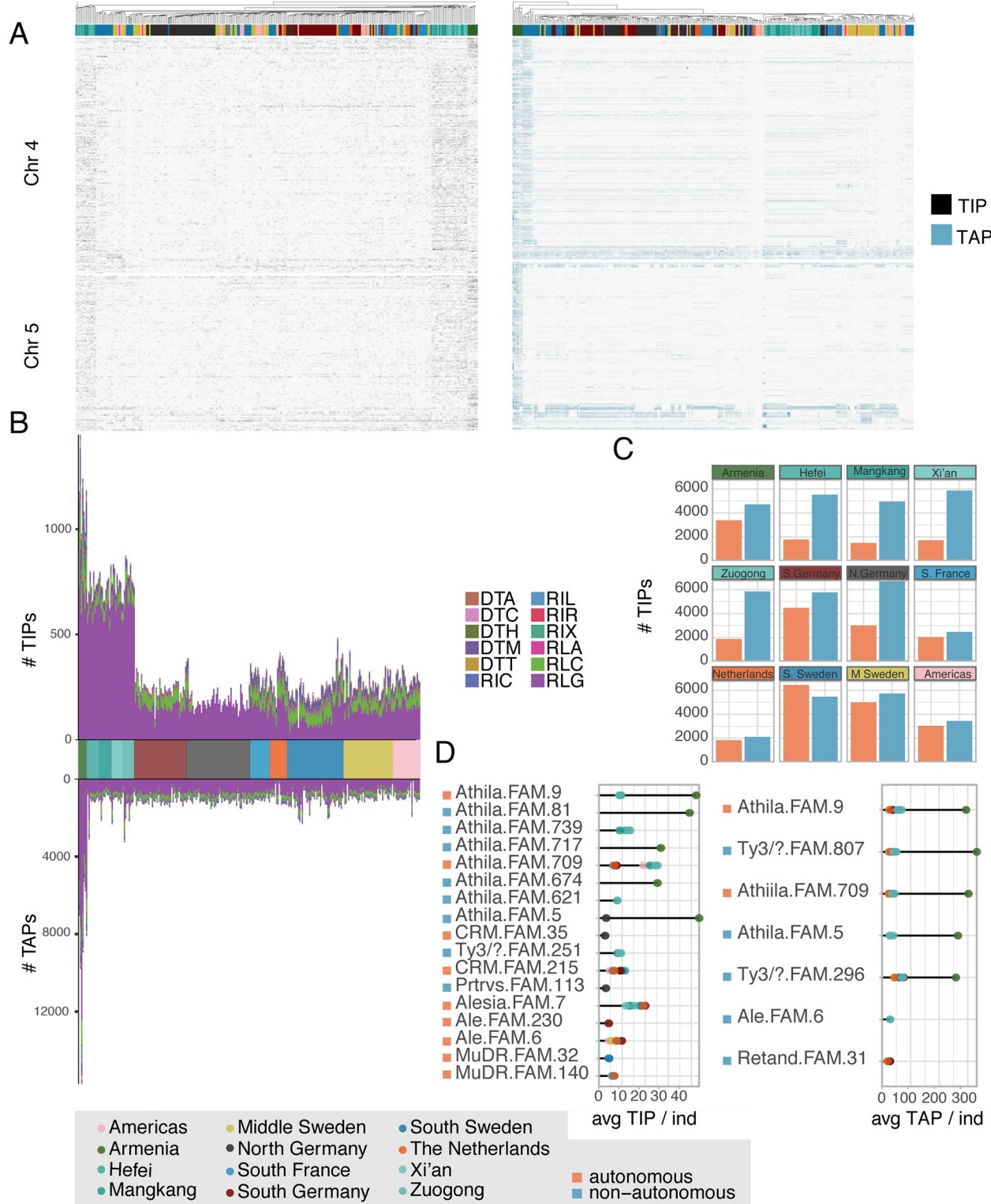

**Fig 3. The *T. arvense* mobilome. (A)** Genomic distribution of TIPs and TAPs in chromosomes 4 and 5, where we observe major TIP/TAP hotspots. TIPs and TAPs along the other chromosomes are shown in S6 Fig. **(B)** Contribution of different superfamilies to transposon insertion polymorphisms (TIPs) and transposon absence polymorphisms (TAPs). **(C)** Frequencies of autonomous and non-autonomous TE-derived TIPs in different geographic regions. **(D)** Average count of TIPs per individual for the five TE families with the highest contribution to either TIPs or TAPs in each geographic region. For all figure panels, the gray box illustrates the color scheme for the geographical populations and for autonomous/non-autonomous families.

## Host control of TE mobility

In *A. thaliana*, natural genetic variation affects TE mobility and genome-wide patterns of TE distribution, driven by functional changes in key epigenetic regulators [14,30–32]. The rich inventory of TE polymorphisms in *T. arvense* offered an opportunity to investigate the genetic basis of TE mobility in a species with a more complex TE landscape. We tested for genome-wide association (GWA) between genetic variants (SNPs and short indels) and TIP load of different TE classes, TE orders and TE superfamilies [4]. We found several GWA hits next to genes that are known to affect TE activity or are good candidates for being involved in TE regulation (Fig 4A–4D). The results differed strongly between class I and class II TEs: while class I TEs were associated with a wide range of genes encoding mostly components of the DNA methylation machinery (Fig 4A–4D), class II TEs were mostly associated with allelic variation at an ortholog of *O. sativa HEAT SHOCK PROTEIN 19* (*HSP19*). Only class I TE superfamilies were enriched for significant associations close to DNA methylation machinery genes (Fig 4B), and this difference was consistent for most superfamilies that belonged to either class I or class II (S7 Fig). The most prominent hits for class I TIPs were near orthologs of *A. thaliana BROMODOMAIN AND ATPase DOMAIN-CONTAINING PROTEIN 1* (*BRAT1*), which prevents transcriptional silencing and promotes DNA demethylation [7], and components of the RNA-directed DNA methylation machinery such as *DOMAINS REARRANGED METHYLTRANSFERASE 1* (*DRM1*), *ARGONAUTE PROTEIN 9* (*AGO9*) and *DICER LIKE PROTEIN 4* (*DCL4*) [33] (Figs 4A–4D, S7 and S8). Another category of genes that emerged in our GWA are genes encoding DNA and RNA helicases such as *RECQL1* and *2* (Figs 4 and S8). Some of our GWA peaks extend over several genes and might reflect associations with less well characterized genes, but others have the strongest associations in individual genes such as *HSP19* and *BRAT1* (S8 Fig). For *HSP19*, the top SNPs are located in introns and it is difficult to predict their effect. *BRAT1* has two highly significant, fully linked SNPs in exons 1 and 4. The SNP in exon 4 (Chr1:63627484) introduces a stop codon that removes part of the ATPase domain and the entire chromatin binding bromodomain, and this mutation almost certainly completely eliminates BRAT1's anti-silencing activity [7].

Since accessions that diverged earlier from the reference had potentially more time to accumulate TIPs, we also estimated the age of all insertions [14] and repeated the GWA using only TIPs younger than 500,000 years. The results were similar to using all TIPs, suggesting that this potential reference bias is unlikely to drive any of the identified associations (S9 Fig).

To further confirm the association between the DNA methylation pathway and class I TE polymorphisms, we used published bisulfite sequencing data to quantify methylation levels of the neighboring regions of TIPs [28]. In all three epigenetic contexts (CG, CHG, CHH; where H stands for all three nucleotides but G), we found a significant increase of methylation up to 1 kb around class I, but not around class II TE insertions (Fig 4E). Taken together, we interpret these results such that class I TE mobility is primarily controlled by the DNA methylation machinery, leading to RdDM spreading around novel insertions, thus creating substantial epigenetic variation beyond TE loci.

## An autonomous Alesia LTR family with insertion preference for specific genomic regions

Our characterization of the *T. arvense* mobilome revealed a strikingly uneven distribution of one autonomous LTR Ty1 family belonging to the Alesia lineage, Alesia.FAM.7. This family encompasses 144 elements in the reference genome, 51 of which are complete copies. Despite being a relatively small TE family, 44 copies are close to genes (< 1 kb), of those, 8 copies are within genes (S3 Table). Across all 4,215 Alesia.FAM.7 TIPs, that is insertions not present in

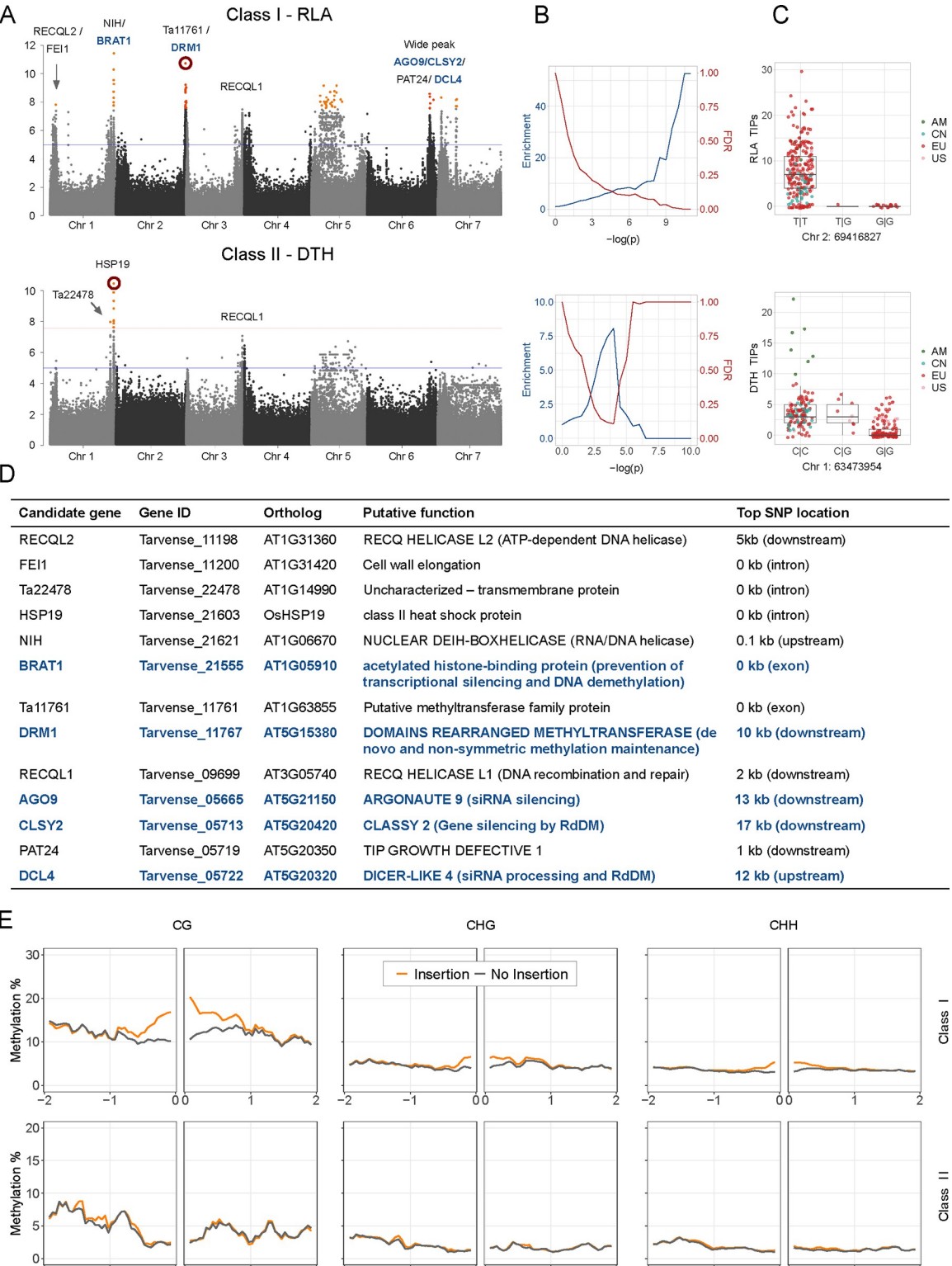

**Fig 4. GWA analysis for TIP load of a class I and a class II TE superfamily.** Results including all superfamilies are shown in S7 Fig. (**A**) Manhattan plots with candidate genes indicated next to neighboring variants. The red line corresponds to a genome-wide significance with full Bonferroni correction, the blue line to a more generous threshold of –log(p) = 5. (**B**) Enrichment and expected FDR of *a priori* candidate DNA methylation machinery genes, for stepwise significance thresholds [28,34]. (**C**) Shown are the allelic

effects of the red-circled variants from the corresponding Manhattan plots on the left. (**D**) Shown are the candidate genes marked in **A**, their putative functions and distances to the top variant of the neighboring peaks. Blue font denotes DNA methylation machinery genes included in the enrichment analyses. (**E**) DNA methylation around class I and class II TIPs in carrier vs. non-carrier individuals.

the reference genome, we found a strong enrichment nearby and within genes, which was the case for ~75% of all insertions (Fig 5A and 5B). The genes potentially affected by these insertions were involved in a wide range of functions, including metabolism and responses to biotic and abiotic factors (Fig 5C). Reference insertions were rarely missing in other accessions, except an intronic reference insertion that was detected as absent in some Swedish accessions. The prevalence of Alesia.FAM.7 TIPs near genes suggests that the skewed distribution in the reference is not so much due to removal of insertions in other regions, but that it reflects an unusual insertion site preference of this family across all examined accessions.

Alesia.FAM.7 is highly similar to the Terestra TE family, first described in *A. lyrata* [35]. The Terestra family, which has been reported in six Brassicaceae, is heat responsive due to a transcription factor binding motif also found in *A. thaliana* ONSEN, where it can be bound by heat shock factor A (HSFA2) via a cluster of four nGAAn motifs called heat responsive elements (HRE) [12]. In Alesia.FAM.7, we found a similar four-nGAAn motif cluster in most copies in the 5' LTR portion of the elements (Fig 5D). A search against the NCBI NT database [36] revealed the presence of this TE family, with an Alesia-diagnostic reverse transcriptase sequence signature, in several additional Brassicaceae (Fig 5E), notably *B. rapa*, *B. napus*, *B. oleracea*, *Raphanus sativus*, and other *Arabidopsis* species, but not in *A. thaliana*. It is conceivable that this heat-responsive, euchromatophilic Alesia family rewires gene regulatory networks between and within Brassicaceae species. We conducted a similar search of a subset of TE families against the NCBI NT database (S10 Fig) and Alesia.FAM.7 was indeed the only deeply conserved TE family with evidence for recent activity.

## Discussion

Although *A. thaliana* and *T. arvense* are close relatives, with evolutionary divergence estimates of 15–24 million years ago [27] and similar life histories in terms of demographic dynamics, geographic expansion, and niche adaptation [25,37], their genomes are very different, one key difference being the significantly higher TE load of the *T. arvense* genome. Exploring the diversity and dynamics of mobile elements in such TE-rich genomes enables a better understanding of the evolution of genome architecture. Here, we report how TEs drive genome variation in *T. arvense* by analyzing the diversity and phylogenetic relationships of TEs, as well as their autonomous status, ongoing activity, and contrasts between biogeographic populations.

Many recent studies have confirmed that several TE families do not insert randomly in the genome, and that their apparent enrichment in specific portions of the genome, such as centromeres, is not simply due to purifying selection [38]. Many TEs have clear insertion site preference [39], both driven by primary DNA sequence and by epigenetic marks, e.g. Ty1 insertions in *A. thaliana* are biased towards regions enriched in H2A.Z [40]. Our results confirm this view whereby the phylogenetic nature of an LTR element plays a role in the observable genome-wide insertion pattern in *T. arvense*. Within the Ty1 elements, Ale elements are preferentially centrophilic whereas Alesia elements are enriched in the genic regions of the genome. For the Ty3 elements, The Retand clade does not show any particular preference across the chromosome, while CRM are centrophilic and Athila insertions are often found in pericentromeric regions. Thus, a phylogenetic classification of TEs, alongside the classification into autonomous and non-autonomous elements, is key to understanding TE dynamics, especially in LTR retrotransposon-rich genomes.

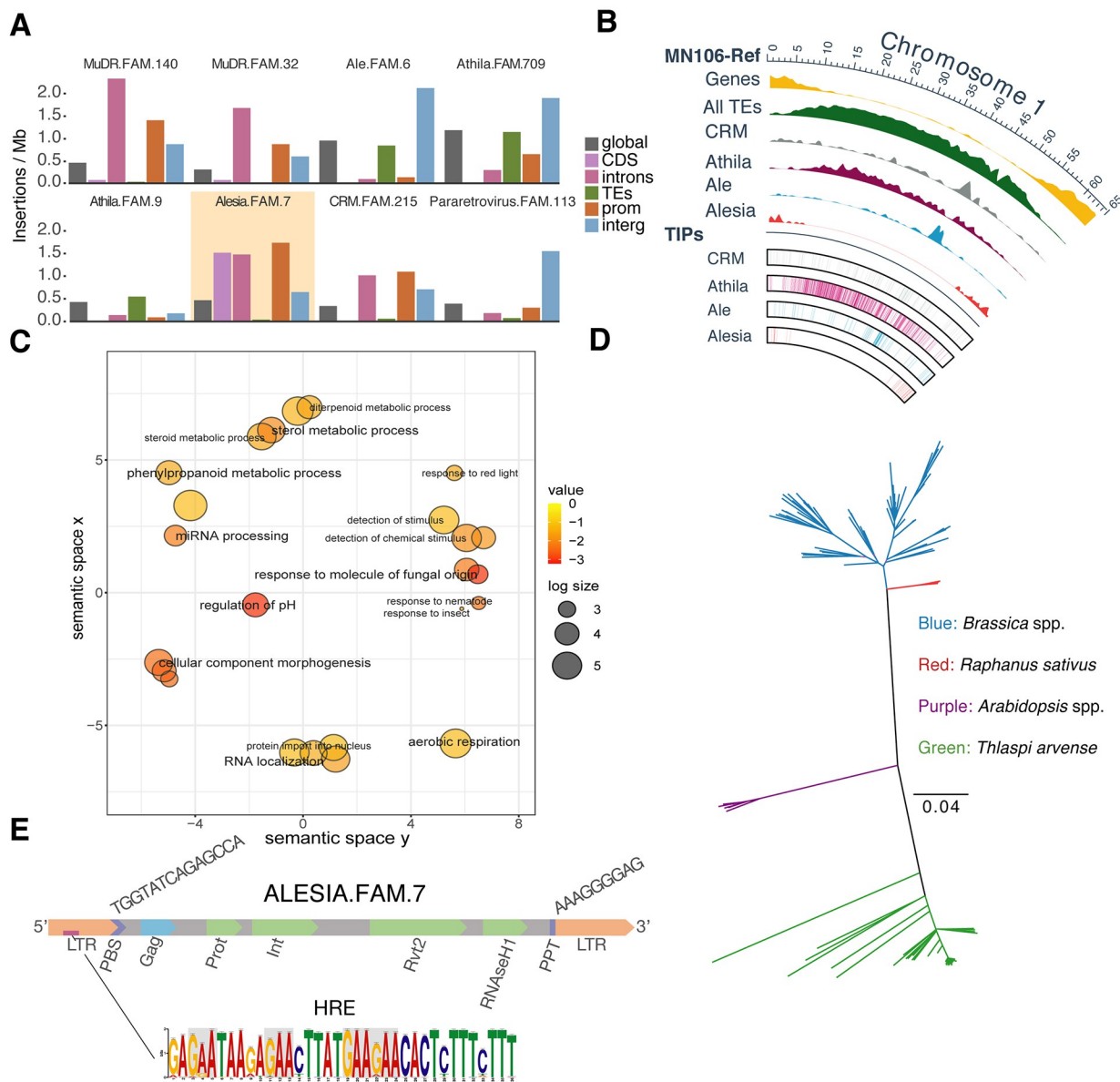

**Fig 5. Summary statistics and characterization of the Alesia.FAM.7 family in *T. arvense* and other Brassicaceae. (A)** Distribution of several TE families across different genomic contexts in *T. arvense* accessions. While several other families, such as MuDR.FAM.140 or CRM.FAM.215, are also often found in introns, Alesia. FAM.7 is the only family that is commonly inserted in coding sequences. **(B)** Distribution of several LTR lineages along chromosome 1 in MN106-Ref. **(C)** GO enrichment of genes associated with Alesia.FAM.7 TIPs. **(D)** Phylogenetic tree of Alesia.FAM.7 related copies across different Brassicaceae. **(E)** Structure of the Alesia. FAM.7 model: 5' Long terminal repeat (LTR); primer binding site (PBS), a tRNA binding site, in this case complementary to *A. thaliana* methionine tRNA; *Gag* domain; *Pol* domains: Protease (Prot), Integrase (Int) and the two subdomains of the reverse transcriptase, the DNA polymerase subdomain (Rvt2) and the RNase H subdomain (RNAseH1); polypurine tract (PPT). The location of a putative heat responsive element (HRE) with the four-nGAAn motif in the LTR is indicated by a purple segment.

We learned that one third of the *T. arvense* genome consists of Ty3/Athila LTR-TEs, which is considerably more than in other Brassicaceae, such as *A. thaliana* and *Capsella rubella*, where Ty1/Ale elements are the most abundant TE lineage [41]. This suggests that a single or multiple ancient Athila bursts may underlie genome size expansion in *T. arvense*. This is in line with the expansion of the Ty3 LTR-TE superfamily, to which Athila belongs, in *Eutrema salsugineum* [42], from which *T. arvense* diverged 10–15 million years ago [43]. Similar Ty3

associated expansions have been reported, for example, for *Capsicum annuum* (hot pepper) [44].

Having established substantial variation in TE content among natural accessions, we asked whether there is also genetic variation for control of TE mobility, as is the case for *A. thaliana* [14,30,31]. Perhaps not too surprisingly, the sets of genes associated with TE mobilization appear to depend on the nature of the TE transposition mechanism. While variation in retrotransposon insertions was strongly associated with several genes involved in the DNA methylation machinery, DNA transposon insertions were instead associated with a single *Heat Shock Protein 19* (*HSP19*) gene, and this was consistent across different class I (retrotransposon) and class II (DNA transposon) superfamilies. Although studies in *A. thaliana* have highlighted differences in the genetic control of methylation and mobility of the two classes of transposons, GWA for CHH methylation of TE families did not produce very different signals for class I and II families [45]. The same was true for TIP-counts of different families and superfamilies as phenotypes [14,32]. Since *HSP19* is an ortholog of an *O. sativa* gene that is absent from the *A. thaliana* reference genome, it is possible that this gene is providing new functionality in *T. arvense*. What this functionality might be is difficult to answer with our data, but different types of HSPs are involved in DNA methylation-dependent silencing of genes and TEs in *A. thaliana* [46], and in controlling transposition in several other organisms [47–49]. That class I and class II TEs in *T. arvense* apparently differ in their genetic requirements for silencing can be potentially linked to our observation that DNA methylation spreads more rapidly from class I than class II TE insertions in this species.

The contrast between Alesia and Athila lineages suggests that TEs may be more than detrimental genome parasites. There are many examples from animals and plants of both TE proteins and TEs themselves having been domesticated and thereby enriching genome function [38,50–52]. While parasitic TEs may constitute the majority of TEs within a given species, there can be different life cycle strategies adopted by TEs [53]. With respect to notable TE families in *T. arvense*, Alesia's gain of HREs might provide a unique selection advantage, allowing it to survive more easily in the genome, as long as copy numbers are low, in a relationship with the host that resembles other forms of symbiotic lifestyle. Further research of this enigmatic Alesia lineage, which is found in many angiosperms [41], could enhance our understanding of the different strategies used by TEs to persist over long evolutionary time scales.

Turning to more practical matters, it might be possible to exploit the preference of Alesia. FAM.7, which is conserved in several Brassicaceae species, for genic insertions as a source of fast genic novelty for crop improvement. TE insertions in exons might disrupt genes, while intronic insertions might modulate alternative splicing or reduce the accumulation of correctly spliced transcripts. An example is provided by *A. thaliana* accessions in which an intronic COPIA insertion in the disease resistance gene *RPP7* shifts the balance between full-length and truncated transcripts [54]. It would therefore be useful to determine how easily Alesia. FAM.7 can be mobilized by heat in *T. arvense*, and conversely, whether heat responsiveness might also be a source of unwanted genetic variation in breeding programs.

## Methods

### Dataset summary

For the investigation of *T. arvense* natural genetic variation (TIPs, TAPs, and short variants), we leveraged Illumina short read data from three studies [26,28,43]. The largest survey investigated both genetic and DNA methylation variation in 207 European accessions (13 from the Netherlands, 16 from the South of France, 42 from the South of Germany, 52 from the North of Germany, 48 from the South of Sweden and 40 from Middle Sweden). In addition, we used

data from 39 Chinese accessions (10 each from Xi'an, Zuogong, and Hefei and 9 from MangKang) [43], 21 from the US, and one each from Chile and Canada [26]. For most of the European accessions, Illumina whole-genome bisulfite-sequencing (BS-seq) data were available as well [28] (S3 Table). We used as reference the assembly generated in [26], together with the gene and TE annotation also generated in that study. To visualize the accession locations in the world map, we used free vector and raster map data from naturalearthdata.com. We reinforced this dataset by sequencing 12 different accessions, 7 Armenian and 5 European, using Illumina paired-end 2x150 bp WGS (S3 Table). Briefly, we grew plants in soil, collected fully developed rosette leaves, snap-froze them in liquid nitrogen and disrupted the tissue to frozen powder. We extracted genomic DNA and prepared Illumina libraries as described before [28]. To validate our TIP analysis we also sequenced our samples using long read HiFi PacBio technology for a single Armenian accession (Ames32867/TA_AM_01_01_F3_CC0_M1_1). For the ancestry analysis, we used two assemblies for *Eutrema salsugineum* and *Schrenkiella parvula* (NCBI ID: PRJNA73205; Phytozome genome ID: 574 respectively) as outgroup species.

## TE analysis of the reference genome

To resolve phylogenetic relationships of the LTR-TEs in *T. arvense* using information from a collection of green plants (Viridiplantae) at REXdb [5], and to classify *T. arvense* LTR-TEs into lineages, we used the DANTE pipeline (https://github.com/kavonrtep/dante) and its Viridiplantae v3.0 database. We used a published *T. arvense* TE library [26] as query with default parameters except for "—interruptions", which we set to 10 to reflect the fact that we used as input the consensus TE models and therefore likely have frameshifts and stop codons in these sequences. Using these identified protein domains, we evaluated whether a given TE family is autonomous, i.e., whether it codes for the entire machinery needed for transposition. An LTR retrotransposon family was considered autonomous with the following domains identified by DANTE: retrotranscriptase, RT; capsid related domain, GAG; RNase H, RH; protease, PROT; and integrase, INT. Autonomous non-LTR retrotransposons, LINEs, had to contain: retrotranscriptase, RT. DNA TE families had to contain: transposase, TPase. DNA TEs of the Helitron superfamily had to contain in addition: DNA helicase, HEL.

After classification, we used the inferred amino acid sequences of the retrotranscriptase domains extracted from Ty3 and Ty1 elements identified by DANTE to produce two multiple sequence alignments using MAFFT with standard parameters [55]. Using RAxML [56], we built a set of phylogenetic trees under a JTT + gamma model, with 100 rapid bootstraps to assess the branch reliability of the NJ tree.

Analysis of intact LTR-TEs analysis and estimates of LTR-TE age used LTRpred [57] against the reference genome with default parameters. We correlated the genomic positions of the *de novo* predicted LTR-TEs with those in the annotation using bedtools [58] intersecting with" -f 0.8 -r" parameters.

To analyze the extent of conservation of TE families larger than 2 kb across Brassicaceae, we ran BLASTN [59] against the NCBI NT database [36], June 2022 release. Next, we filtered the result by requiring 80% identity and 80% alignment coverage of the query sequence. For Alesia.FAM.7 TE family filtered matches, we performed a multiple sequence alignment of the remaining matches using MAFFT [55] with default settings and constructed a tree with RaxML [56] with the parameters "-model JTT+G—bs-trees 100". To *de novo* discover nGAAn motifs in all the sequences of Alesia.FAM.7, we ran MEME [60] with the following parameters "-mod zoops -nmotifs 3 -minw 6 -maxw 50 -objfun classic -revcomp -markov_order 0".The *de novo* deemed HRE motif selected had 4 nGAAn clusters in the reverse strand: AAAGAAAGAGTGTTCTTCATAAGTTCTCTTATTCTC (E-value = 2.8e-33).

Expression analysis of reference TEs was performed using TEspeX [61]. We obtained paired-end RNA seq data from 27 samples comprising nine different tissues from the MN106-Ref reference accession [26]. We obtained raw counts for each library by mapping the reads to both transcripts of protein coding genes and to the TE consensus library. Raw counts were normalized as suggested [61] (RPM: raw counts/total mapped reads x 1 million). We used a non-parametric Wilcoxon rank-sum test to compare expression between autonomous and non-autonomous TE families.

## Short variant calling

We called variants with GATK4 [62], following best practices for germline short variant discovery (https://gatk.broadinstitute.org/hc/en-us/sections/360007226651-Best-Practices-Workflows), as described in [28]. Briefly, we trimmed reads, removed adaptors, and filtered low quality bases and short reads ($\leq$25 bp) using cutadapt v2.6 [63]. We aligned trimmed reads to the reference genome [26] with BWA-MEM v0.7.17 [64], marked duplicates with *MarkDuplicatesSpark* and ran *Haplotypecaller*, generating GVCF files for each accession. To combine GVCF files, we ran *GenomicsDBImport* and *GenotypeGVCFs* successively for each scaffold, and then merged files with *GatherVcfs*, to obtain a multisample VCF file. Based on quality parameters distributions, we removed low-quality variants using *VariantFiltration* with specific parameters for SNPs (QD < 2.0 || SOR > 4.0 || FS > 60.0 || MQ < 20.0 || MQRankSum < -12.5 || ReadPosRankSum < -8.0) and other variants (QD < 2.0 || QUAL < 30.0 || FS > 200.0 || ReadPosRankSum < -20.0). We filtered variants with *vcftools* v0.1.16 [65], retaining only biallelic variants with at most 10% missing genotype calls, and Minor Allele Frequency (MAF) > 0.01. Finally, we imputed missing genotype calls with *BEAGLE* 5.1 [66], obtaining a complete multisample VCF file. All the code for short variants calling, filtering and imputation can be found on GitHub (https://github.com/Dario-Galanti/BinAC_varcalling).

For calculating site frequency spectra, we used all biallelic SNPs with Minor Allele Count (MAC) of at least two. To assess the population structure of our dataset, we pruned variants in strong LD using *PLINK* [67] with the following parameters "—indep-pairwise 50 5 0.8" and then ran PCA analyses to assess the variance of natural variation. Due to the high divergence of the Armenian accessions from the rest, we ran separate PCAs with and without these accessions, to highlight the structure of the remaining populations (S4 Fig).

Lastly, we analyzed the genetic relatedness among accessions from different geographic regions constructing a maximum likelihood tree using *TREEMIX* [68] with 2,500 bootstrap replicates without considering migration flow and using as an outgroup two sister species, *Eutrema salsugineum* and *Schrenkiella parvula*. We merged all 2,500 independent *treemix* runs and generated a consensus tree with the *Phylip* "consense" command (https://evolution.genetics.washington.edu/phylip/).

## TE polymorphism calling

To identify TE insertion polymorphisms (TIPs), we used *SPLITREADER* [32] as described in [69]. We applied two custom steps (https://github.com/acontrerasg/Tarvense_transposon_dynamics). In short, we removed Helitron insertions, as they have been shown to have a high false positive ratio [32].

Next, we mapped short reads from the reference accession MN106 to the reference genome to identify regions of aberrant coverage, using sample SAMEA9464759 from ENA project PRJEB46635 [26]. We calculated read coverage (RC) in 100 bp windows, adjusted for GC content [70], and excluded windows with abnormal coverage, arbitrarily defined as threefold

lower or higher than the genome wide, GC content adjusted mean. Any TIPs in these regions, which corresponded to ~16% of the reference genome, were excluded from the final dataset. Lastly, we removed TIPs with >100 reads 500 bp upstream and/or downstream of the TIP, because this suggested aberrant structural variants in the sample, not reflected in the reference. To calculate the variant frequency spectra of TIPs, we classified TIPs as shared between two or more accessions if coordinates were identical. To estimate the age of insertions, we used this same classification and calculated the maximum pairwise divergence (number of SNPs) between each combination of two carriers, in the 70 kb region around the insertion [14], using simply the number of private SNPs for singletons. We then extrapolated the most likely age based on *A. thaliana* mutation rate [71], assuming 1 generation per year.

To detect TIPs using *SPLITREADER*, a collection of TEs is required. We used a representative subset of the total number of TEs present in the *T. arvense* reference genome, generated with a custom script. As a selection criterion, we defined representatives according to the consensus TE sequence of each family and the five longest individual members of each family. If a family consisted of < 5 members, all members were used.

We visually inspected 2,790 TIPs spanning all analyzed TE superfamilies and all accessions using a visual browser. Over 70% of TIPs were deemed correct, which is in line with reports from other studies in *A. thaliana* [32] and tomato [72].

To further confirm our TIPs, we generated HiFi PacBio long reads for an Armenian accession (Ames32867/TA_AM_01_01_F3_CC0_M1_1). We stratified seeds at 4˚C for one month and germinated them on soil. One month after germination, we subjected plants to 24h dark prior to harvesting. We extracted high molecular weight (Hmw) DNA as described [73] using 600 mg of ground rosette material. Using a gTUBE (Covaris) we sheared 10 μg of HMW DNA to an average fragment size of of 24 kb and prepared two independent non-barcoded HiFi SMRTbell libraries using SMRTbell Express Template Prep Kit 2.0 (PacBio). We pooled the two libraries and performed size-selection with a BluePippin (SageScience) instrument with 10 kb cutoff in a 0.75% DF Marker S1 High Pass 6–10 kb v3 gel cassette (Biozym). We sequenced the library on a single SMRT Cell (30 hours movie time) with the Sequel II system (PacBio) using the Binding Kit 2.0. Using PacBio CCS with "—all" mode (https://ccs.how/), we generated HiFi reads (sum = 31 Gb, n = 1,633,975, average = 19 kb). We called structural variants (SVs) against the reference using *Sniffles2* [74]. 71% of the TIPs called in this accession using short reads had a PacBio HiFi-read supported SV within 200 bp, in line with our visual assessment of TIP quality.

Using paired-end short read Illumina data, we also screened for TE absence polymorphisms (TAPs). First, we calculated the GC-corrected median read depth (RD) in genome-wide 10 bp bins for short-read data sets from all accessions and from two reference controls. For every annotated TE ≥ 300 bp, we extracted its corresponding RD-bins for both the controls and a single sample and used a non-parametric test (Wilcoxon Rank Sum) to compare the bins of the focal sample with the bins of both controls. If i) the annotated TE showed a significant difference in coverage between the focal accession and the mean of the controls, and ii) the median coverage of that TE showed at least a 10-fold reduction in the focal accession compared to the all accession median coverage, then such a TE was considered absent in the focal accession. To exclude the possibility that our TAP calls were the result of major rearrangements in the vicinity of the TAP call, we calculated the coverage of the flanking regions of the TAPs and removed those with < 5X or > 50X mean coverage.

## Genome-wide Association between TE polymorphisms and genomic regions

To detect genetic variants associated with variation in TE content, we ran GWA using the number of TIPs of different classes, orders and superfamilies as phenotypes. We used mixed

models implemented in *GEMMA* [75], correcting for population structure with an Isolation-By-State (IBS) matrix. Starting from the complete VCF file obtained from variant calling, we used *PLINK* [67] to prune SNPs in strong LD (—indep-pairwise 50 5 0.8) and computed the IBS matrix. We tested for associations between TIP counts and all variants with MAF > 0.04 (SNPs and short INDELs). We log-transformed TIP counts to approximate a log-normal distribution of the phenotype. To quantify the potential effects of components of the epigenetic machinery on TE content, we calculated the enrichment of associations in the proximity of a custom list of genes with connections to epigenetic processes [28] for increasing cutoffs [34]. Briefly, we assigned an "*a priori* candidate" status to all variants within 20 kb of the genes from the list and calculated the expected frequency as the fraction between "*a priori* candidate" and total variants. We calculated enrichment for -log(p) threshold increments, comparing the fraction of significant *a priori* candidates (observed frequency) to the expected frequency. We further calculated the expected upper bound for the false discovery rate (FDR) as described in [34]. The code to run GWA and the described enrichment analysis is available on GitHub (https://github.com/Dario-Galanti/multipheno_GWAS/tree/main/gemmaGWAS).

### DNA methylation around insertions

To investigate cytosine methylation in the proximity of TIPs, we leveraged Whole Genome Bisulfite Sequencing (WGBS) data from the European accessions, using multisample unionbed files [28]. To reduce technical noise, we first excluded singleton TIPs and TiPs within 2 kb of another TIP or 1 kb to annotated TEs. We calculated average methylation of accessions with and without a focal TIP in 2 kb flanking regions. We then combined methylation values of all TIPs in 50 bp bins of the 2 kb flanking regions, averaging all positions within each bin. Finally, we calculated the moving average (arithmetic mean) of 3 bins to smooth the curves. The workflow was based on custom bash and python scripts available at https://github.com/acontrerasg/Tarvense_transposon_dynamics.

### Intersection with genomic features and Gene Ontology enrichment analysis

To investigate the targeting behavior of different TE families or superfamilies, we counted TIPs in different genomic features with *bedtools* [58] and divided them by the total genome space covered by each feature to obtain relative insertion density. We turned to gene ontology (GO) enrichment analysis to characterize genes potentially affected by insertions, using all genes located within 2 kb of an insertion. Briefly, we extracted GO terms from the *T. arvense* annotation and integrated them with the terms from *A. thaliana* orthologs identified by *OrthoFinder2* [76]. We assessed enrichment with *clusterProfiler* [77] and piped all terms with p value < 0.05 to *REVIGO* [78], using default parameters.

### Code availability

Code used for analysis and figures can be found at: https://github.com/acontrerasg/Tarvense_transposon_dynamics.

### Supporting information

**S1 Table. Summary statistics of previously annotated TEs for the *T. arvense* reference genome MN106-Ref.**
(XLSX)

**S2 Table. Lineages of LTR-TEs in the *T. arvense* genome MN106-Ref.**
(XLSX)

**S3 Table. List of datasets that were uploaded to the Zenodo repository: https://doi.org/10.5281/zenodo.10161730 (10.5281/zenodo.6372331).**
(XLSX)

**S1 Fig. Comparison of autonomous and non-autonomous TE families in *T. arvense* MN106-Ref. (A)** Absolute (left) and relative (right) fraction of autonomous and non-autonomous elements in each TE superfamily. (**B**) Comparison of the fraction of autonomous and non-autonomous elements in each TE superfamily (left). Size comparison of the TE copies according to their autonomy per superfamily (right). (**C**) Contribution of each superfamily and their autonomous/non-autonomous fraction to total genome size in Mb. (**D)** Distribution of size and copy number per LTR retrotransposon lineage. (**E**) TE expression in autonomous vs. non-autonomous TEs.
(TIF)

**S2 Fig. SNP-based maximum likelihood tree of *T. arvense* populations.** Based on a model without migration, 2,500 bootstraps. Node weights represent bootstrap values. Outgroup species at the bottom.
(TIF)

**S3 Fig. Frequency distribution of TIPs overlapping with annotated genes and TEs.** TIP allele frequencies near other TEs are significantly lower than near genes (Wilcoxon Rank Sum test, $p < 2.22E^{-16}$).
(TIF)

**S4 Fig. SNP-based PCA of a subset of *T. arvense* accessions.** The Armenian accessions, which are outliers in the PCA using all accessions (Fig 2), were excluded from this new PCA analysis, which shows how Chinese and European accessions cluster separately. We also observe part of the south Sweden accessions clustering far from the rest of the European accessions.
(TIF)

**S5 Fig. PCA analysis of 279 individuals of *T. arvense*.** A presence/absence matrix of either TIPs (left) or TAPs, (right) was used as input to calculate PCA. This result recapitulates the clustering pattern observed with the SNP-PCA.
(TIF)

**S6 Fig. Genomic distribution of TIPs and TAPs along all seven chromosomes of *T. arvense*.** Color columns indicate to which biogeographical population each accession belongs to.
(TIF)

**S7 Fig. Complete GWA results for TIP load.** Left: Manhattan plots for each TIP superfamily load. The genome-wide significance (red line) corresponds to a full Bonferroni correction, the suggestive line (blue) to a more generous hard threshold of –log(p) = 5. Genes next to top variants are labeled with names, blue font indicates genes with link to DNA methylation included in the enrichment analyses. Middle: Enrichment and expected FDR of genes with link to DNA methylation, for significance threshold increments [28,34]. Right: QQplots of p-values.
(TIF)

**S8 Fig. Zoom-in of GWA peaks with candidate genes highlighted.** The genome-wide significance (dotted red line) corresponds to a full Bonferroni correction. DNA methylation machinery genes used for the enrichment of *a priori* candidates are depicted in blue, other genes that might affect transposition in red. The putative knock-out SNP disrupting the function of

*BRAT1* is depicted in green.
(TIF)

**S9 Fig. GWA results for genome-wide load of TIPs younger than 500,000 years.** Left: Manhattan plots for load of TIPs for each TE superfamily. The genome-wide significance (red line) corresponds to a full Bonferroni correction, the suggestive line (blue) to a more generous hard threshold of $-\log(p) = 5$. Genes next to top variants are labeled with names, blue font indicates genes with link to DNA methylation included in the enrichment analyses. Middle: Enrichment and expected FDR of genes with links to DNA methylation, for significance threshold increments. Right: QQplots of p-values.
(TIF)

**S10 Fig. BLASTN hits of *T. arvense* TE families with model sizes $> 4$ kb against the NCBI NT database, June 2022 release.** We filtered the matches using the 80/80/80 rule, and further constrained matches to fulfill $> 2$kb length criteria. The x-axis denotes the number of species with at least 1 hit. Each family has at least one hit, namely *T. arvense* itself. TE families with more than 5 hits are highlighted. The number of TIPs in *T. arvense* populations is shown in parentheses for the highlighted families to indicate that there is no obvious correlation between mobility in *T. arvense* and phylogenetic conservation.
(TIF)

## Acknowledgments

We thank Haim Ashkenazy, Wei Yuan and Gautam Shirsekar for technical advice, Christa Lanz for support during PacBio HiFi library preparation and Alejandra Duque-Jaramillo, Tess Renahan and Rebecca Schwab for comments on the manuscript. We also thank Kevin M. Dorn for sharing *T. arvense* seeds. For computing, we acknowledge Prof. Peter Stadler at the University of Leipzig and David Langenberger from ecSeq, for hosting the EpiDiverse servers, and the High Performance and Cloud Computing Group at the Zentrum für Datenverarbeitung of the University of Tübingen for managing the BinAC server.

## Author Contributions

**Conceptualization:** Adrián Contreras-Garrido, Dario Galanti, Andrea Movilli, Oliver Bossdorf, Hajk-Georg Drost, Detlef Weigel.

**Data curation:** Adrián Contreras-Garrido, Dario Galanti, Andrea Movilli.

**Formal analysis:** Adrián Contreras-Garrido, Dario Galanti.

**Funding acquisition:** Claude Becker, Oliver Bossdorf, Detlef Weigel.

**Investigation:** Adrián Contreras-Garrido, Dario Galanti, Claude Becker, Oliver Bossdorf, Hajk-Georg Drost, Detlef Weigel.

**Methodology:** Adrián Contreras-Garrido, Dario Galanti.

**Project administration:** Oliver Bossdorf, Hajk-Georg Drost, Detlef Weigel.

**Resources:** Adrián Contreras-Garrido, Dario Galanti, Andrea Movilli, Claude Becker, Oliver Bossdorf, Hajk-Georg Drost, Detlef Weigel.

**Software:** Adrián Contreras-Garrido.

**Supervision:** Oliver Bossdorf, Hajk-Georg Drost, Detlef Weigel.

**Validation:** Adrián Contreras-Garrido, Dario Galanti, Andrea Movilli, Hajk-Georg Drost, Detlef Weigel.

**Visualization:** Adrián Contreras-Garrido, Dario Galanti.

**Writing – original draft:** Adrián Contreras-Garrido, Dario Galanti, Claude Becker, Oliver Bossdorf, Hajk-Georg Drost, Detlef Weigel.

**Writing – review & editing:** Adrián Contreras-Garrido, Dario Galanti, Andrea Movilli, Claude Becker, Oliver Bossdorf, Hajk-Georg Drost, Detlef Weigel.

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
