## [Decision Letter · Decision Letter 0]

26 Oct 2023

Dear Dr Weigel,

Thank you very much for submitting your Research Article entitled 'Transposon dynamics in natural populations of the emerging oilseed crop Thlaspi arvense' to PLOS Genetics.

The manuscript was fully evaluated at the editorial level and by three independent peer reviewers. The reviewers appreciated the attention to an important topic but identified some concerns that we ask you address in a revised manuscript.

We therefore ask you to modify the manuscript according to the review recommendations. Your revisions should address the specific points made by each reviewer.

Yours sincerely,

Yalong Guo, Ph.D.

Guest Editor

PLOS Genetics

Li-Jia Qu

Section Editor

PLOS Genetics

Reviewer's Responses to Questions

**Comments to the Authors:**

Reviewer #1: Contreras-Garrido et al. present a comprehensive characterization of transposable element (TE) insertion polymorphisms in Thlaspi arvense. Utilizing resequencing data from 280 accessions worldwide, they report extensive variation in the number of polymorphic TE insertions among accessions, with Armenian accessions displaying the highest number of polymorphic insertions. Their exploration of the genetic architecture of this variation through GWAS, using the number of polymorphic TE insertions as a trait, reveals associations with several genes involved in the epigenetic control of TEs. Notably, a recurrent association with a gene encoding a poorly characterized HSP is uncovered. These results are of particular significance, as similar studies in A. thaliana have identified similar epigenetic pathways but through different genes, suggesting species-specific genetic architectures for TE mobilization. Additionally, the identification of a recently expanded Ty1/Copia retrotransposon in Thlaspi arvense, preferentially accumulated within or nearby genes, and containing putative heat-responsive elements analogous to those in the heat-responsive ONSEN retrotransposon from A. thaliana, adds another layer of insight.

The work is sound and meticulously executed, especially given the challenges of identifying TE insertion polymorphisms using short-reads. The identification of recently active TE families absent in the model species A. thaliana, as well as the multiple GWAS findings, are intriguing and pertinent contributions to our comprehension of TE evolution, control mechanisms, and their influence on genome dynamics. Furthermore, the catalog of polymorphisms established here can serve as a valuable resource for researchers and breeders interested in the emerging crop Thlaspi arvense.

I have nonetheless some technical concerns:

1- The observation of a higher load of high-frequency TIPs over gene-rich regions, as noted by the authors on page 8, is unexpected. This conclusion is drawn from Figure 2E; however, this plot may be misleading. Outliers like the few almost fixed TE insertions near chromosome ends can create the impression of an overall trend. It may be more informative to display the average allele frequency by genomic windows (e.g. 100/500Kb windows) instead. Furthermore, the challenge in detecting polymorphic insertions within TE-rich regions compared to unique sequences could skew the allele frequency spectrum (AFS) towards lower values in the former. One possibility would be determining the mappability of Illumina short-reads at each insertion site to distinguish between “true absences” of TE insertion polymorphisms from a “lack of informative” reads. The latter should be considered as "NA" rather than "absences."

2- GWAS were conducted on TIPs, which are defined as non-reference insertions in this work. Since the number of TIPs is expected to correlate with the genetic distance to the reference genome, as suggested in Figure 3B, the GWAS might primarily capture this reference bias. To mitigate such bias, it might be desirable to employ the number of rare TIPs, as well as common TAPs, per accession as a proxy for TE mobilization. These numbers are expected to represent the most recent non-reference and reference insertions, respectively.

Minor comments:

Page 3. “In the Ty3 superfamily, Ale elements…”. Ale does not belong to Ty3 superfamily.

Page 14. “Although studies in A. thaliana have highlighted differences…., they are not as striking compared to the evidence we found here”. I am not sure I follow the reasoning behind why the findings presented here are considered more striking. Please provide further elaboration on this argument.

Page 14. “…differences in the genetic control of silencing of class I and class II TEs is further supported by … DNA methylation spreads from class I TE insertions, but not from class II TE insertions” . It is unclear from this work how methylation spreading relates to the differential control of TEs. Please place these results in the context of current knowledge regarding DNA methylation spreading and the various epigenetic pathways involved.

Page 14. “Modulation of gene expression via intronic insertions”. It is unclear how intronic insertions can modulate gene expression. Please expand on this point and incorporate relevant literature.

Reviewer #2: The manuscript by Contreras-Garrido & Galanti and coauthors explores the transposable element landscape in natural populations of Thlaspi arvense across different geographic regions. The authors classified the TEs and their genetic variability within the samples/populations and explored the meaning of the variability. The findings are intriguing and contribute to the knowledge of the emerging oilseed crop Thlaspi arvense. Although the authors used GWA to analyze TE insertion polymorphisms in 280 wild accessions, other datasets would improve the manuscript, allowing important conclusions to be drawn.

1. Pg 3 - …ongoing mobilization activity (13,18,56). Is unclear what those numbers mean. Could the authors clarify?

2. Fig 1B – A color key in the panel (not only in the legend) will help the reader understand the circular plot better/faster.

3. Transcriptome (or other expression analysis) would help to address if the autonomous TEs are active or silenced.

4. Does the methylation analysis not include TEs-body, only the surrounding regions? Are the TEs bodies methylated? Which tissue was used for the WGBS?

5. Is tempting and makes sense that RdDM would be responsible for the methylation, however, more information is necessary to conclude that. Does the TEs generate mRNA and sRNAs? Other silencing mechanisms are possible, for instance, alternative TEs silencing mechanisms such as in Habu et al., 2006 and Rangwala & Richards, 2007.

6. The common name (“penny cress”) should be used in the abstract

Reviewer #3: Transposable elements (TEs) account for a large proportion of eukaryotic genomes and contribute to evolution of traits through generating new genic or regulatory sequences. However, our understanding on how and to what extent transposable elements may contribute to trait evolution in a given species are limited. In the current manuscript, the authors analyzed 280 wild accessions of an emerging oil crop, Thlaspi arvense, to obtain a map of TE variations in this population. Thlaspi arvense is a member of the Brassicaceae family and its genome contains a large proportion of TE compared to most of other Brassicaceae species with genome sequences available. They performed a comprehensive analysis on TE diversity in Thlaspi arvense genome and in the wild population. The new findings are a new Ty1/Alesia lineage that closely related to protein-coding genes and new host factors associated with TE activity. The manuscript is well organized and well written. I enjoyed reading the manuscript. I summarized the strength and weakness of the manuscript that may help the authors to improve the manuscript.

Strength:

TE activity might be very difference from species to species. Therefore, a new model for studying TE activity and TE regulation is always attractive. The authors performed a comprehensive analysis of TE polymorphism and identified host factors associated with TE regulation in 280 wild accessions of the emerging old crop, Thlaspi arvense. Overall, the results suggest that TE regulation in Thlaspi arvense is different from Arabidopsis thaliana. This warrants further studies to explore these host factors and impact of TEs on host adaption.

Weakness:

The authors reported several interesting observations. However, the analysis could be extended to provide detailed evidence to support the conclusion. For example, the authors identified several candidate genes associated with TE activity. In Figure S8, the authors showed GWAS signals, which suggest that multiple genes were included in the peak region. The authors could provide detailed analysis on how they determine the candidate genes using haplotype or gene expression analysis. Similarly, the authors claimed heat responsive elements in Alesia.FAM.7. Instead of stating that “It is conceivable that this heat-responsive, euchromatophilic Alesia family rewires gene regulatory networks between and within Brassicaceae species.” The authors could setup an experiment to prove Alesia.FAM.7 transcription can be induced upon heat stress.

Minor concerns:

1, In page 3, the authors classified TEs into autonomous TEs and nonautonomous TEs but I did not find the related description in the method section. Please add these details in the method.

2, In page 5, the authors stated that “Because a substantial part of the reference genome consistent of autonomous, like still active, TE families, we want to learn how variable TE content is in a large collection of natural accessions”. First, autonomous TEs do not mean it is active. This is not accurate. The author can safely remove “like still active” in this sentence. Second, the authors may want to say something after “learn how variable TE content is”. Please finish the sentence.

3, In page 9, the authors claimed that “In all three epigenetic contexts (CG, CHG, CHH), we found a significant increase of methylation up to 1 kb around class I, but not around class II TE insertions”. Although the spreading of DNA methylation by class I TE has been reported, further analysis could be performed to provide some case examples of methylation spreading caused by these new TE insertions.

**Have all data underlying the figures and results presented in the manuscript been provided?**

Reviewer #1: Yes

Reviewer #2: Yes

Reviewer #3: **No: **The record is available. The related data has not been released yet.

PLOS authors have the option to publish the peer review history of their article (what does this mean?). If published, this will include your full peer review and any attached files.

Reviewer #1: **Yes: **Leandro Quadrana

Reviewer #2: No

Reviewer #3: No

---

## [Decision Letter · Decision Letter 1]

17 Jan 2024

Dear Dr Weigel,

We are pleased to inform you that your manuscript entitled "Transposon dynamics in the emerging oilseed crop Thlaspi arvense" has been editorially accepted for publication in PLOS Genetics. Congratulations!

Yours sincerely,

Yalong Guo, Ph.D.

Guest Editor

PLOS Genetics

Li-Jia Qu

Section Editor

PLOS Genetics

Comments from the reviewers (if applicable):

Reviewer's Responses to Questions

**Comments to the Authors:**

Reviewer #1: The authors addressed all my concerns.

Reviewer #3: Thanks for the opportunity to review the revised of the manuscript. The authors have addressed most of my concerns. The manuscript has been significantly improved.

**Have all data underlying the figures and results presented in the manuscript been provided?**

Reviewer #1: Yes

Reviewer #3: Yes

PLOS authors have the option to publish the peer review history of their article (what does this mean?). If published, this will include your full peer review and any attached files.

Reviewer #1: **Yes: **Leandro Quadrana

Reviewer #3: No

**Data Deposition**

http://datadryad.org/submit?journalID=pgenetics&manu=PGENETICS-D-23-00891R1

**Press Queries**

---

## [Editor Report · Acceptance letter]

26 Jan 2024

PGENETICS-D-23-00891R1 

Transposon dynamics in the emerging oilseed crop Thlaspi arvense 

Dear Dr Weigel, 

We are pleased to inform you that your manuscript entitled "Transposon dynamics in the emerging oilseed crop Thlaspi arvense" has been formally accepted for publication in PLOS Genetics! Your manuscript is now with our production department and you will be notified of the publication date in due course.

With kind regards,

Judit Kozma

PLOS Genetics

On behalf of:
